# A robust method of nuclei isolation for single-cell RNA sequencing of solid tissues from the plant genus *Populus*

**Daniel Conde**[1☯], **Paolo M. Triozzi**[1☯], **Kelly M. Balmant**[1], **Andria L. Doty**[2], **Mariza Miranda**[2], **Anthony Boullosa**[2], **Henry W. Schmidt**[1], **Wendell J. Pereira**[1], **Christopher Dervinis**[1], **Matias Kirst**[1,3]*

**1** School of Forest, Fisheries and Geomatic Sciences, University of Florida, Gainesville, Florida, United States of America, **2** Interdisciplinary Center for Biotechnology, Flow Cytometry & Imaging Core, University of Florida, Gainesville, Florida, United States of America, **3** Genetics Institute, University of Florida, Gainesville, Florida, United States of America

☯ These authors contributed equally to this work.
* mkirst@ufl.edu

**Data Availability Statement:** All relevant data are within the paper and its Supporting information files.

## Abstract

Single-cell transcriptome analysis has been extensively applied in humans and animal models to uncover gene expression heterogeneity between the different cell types of a tissue or an organ. It demonstrated its capability to discover key regulatory elements that determine cell fate during developmental programs. Single-cell analysis requires the isolation and labeling of the messenger RNA (mRNA) derived from each cell. These challenges were primarily addressed in mammals by developing microfluidic-based approaches. For plant species whose cells contain cell walls, these approaches have generally required the generation of isolated protoplasts. Many plant tissues' secondary cell wall hinders enzymatic digestion required for individual protoplast isolation, resulting in an unequal representation of cell types in a protoplast population. This limitation is especially critical for cell types located in the inner layers of a tissue or the inner tissues of an organ. Consequently, single-cell RNA sequencing (scRNA-seq) studies using microfluidic approaches in plants have mainly been restricted to *Arabidopsis* roots, for which well-established procedures of protoplast isolation are available. Here we present a simple alternative approach to generating high-quality protoplasts from plant tissue by characterizing the mRNA extracted from individual nuclei instead of whole cells. We developed the protocol using two different plant materials with varying cellular complexity levels and cell wall structure, *Populus* shoot apices, and more lignified stems. Using the 10× Genomics Chromium technology, we show that this procedure results in intact mRNA isolation and limited leakage, with a broad representation of individual cell transcriptomes.

## Introduction

Single-cell transcriptome analysis has revolutionized genomic research in humans and animal models by allowing cell-specific gene expression programs to be characterized and new cell

**Funding:** This work was supported by the Department of Energy Office of Science Biological and Environmental Research (www.energy.gov/science/ber/biological-and-environmental-research) with a grant (DE-SC0018247) to MK. The funder had no role in study design, data collection and analysis, decision to publish, or preparation of the manuscript.

**Competing interests:** The authors have declared that no competing interests exist.

types to be discovered. Single-cell RNA sequencing (scRNA-seq) has led to the creation of organogenesis cell atlases in mice and humans [1–4]. This approach has made it possible to determine the cellular changes that occur during the development of organs such as the heart and brain [5, 6]. The analysis of individual cell types' developmental trajectory has identified the transcriptional program that determines their differentiation [7]. Moreover, the comparison of single-cell transcriptome programs across species and conditions has made it possible to understand how organs develop differently between mice and humans and how individual cell types respond differently to external stimuli [8].

Although scRNA-seq has rapidly been adopted in animal systems, its application has lagged in plant species. For instance, by the end of 2020, only five single-cell transcriptome studies involving plants had been deposited in the public repository from the European Molecular Biology Laboratory Single-Cell Expression Atlas database (ebi.ac.uk/gxa/sc). Within the same period, over 150 studies with animal species have been made available. The discrepancy in resource availability lies in part in the unique challenges associated with analyzing individual plant cells. While several early methods have been employed for this purpose (e.g., laser capture microdissection), they have been hampered by low throughput. This challenge was addressed in studies with animal species with the development of approaches for separating and labeling mRNAs from individual cells using microfluidics.

For plant species, whose cells contain cell walls, the use of microfluidic approaches for single-cell analysis has generally relied on the generation of protoplasts. For the proper functioning of the microfluidics approaches, suspensions need to contain intact protoplasts that are well dissociated and free of cell debris. Failure to meet these requirements results in low cell count, mRNA cell leakage, and the microfluidics apparatus's obstruction. Unsurprisingly, the existing scRNA-seq studies in plants have been primarily in *Arabidopsis* roots [9–12], for which procedures of cell (protoplast) isolation are well established. However, protoplast preparation is highly dependent on the properties of the tissue in consideration, and methods are not immediately transferable to other tissue types nor across plant species. Cell dissociation by removal of the cell wall also damages sensitive cells while often failing to release others. Finally, enzymatic and mechanical methods for single-cell dissociation have been shown to introduce stress-induced transcriptional artifacts [13, 14]. The generation of damaged cells, non-dissociated cell clusters and abundant cell debris is often incompatible with microfluidic devices that are used for scRNA-seq sample preparation.

The challenge of generating high-quality protoplasts from plant tissue can be partially addressed by taking an alternative approach–to characterize the mRNA extracted from individual nuclei instead of whole cells. In mammals, solid tissues (e.g., kidney) that contain difficult-to-dissociate cell types are problematic to analyze using standard approaches without resulting in RNA degradation, transcriptional stress responses, and cell aggregates. These studies have increasingly adopted the use of RNA-seq analysis of single-nuclei (often referred to as snRNA-seq) to address these limitations. These studies were shown to capture a broader diversity of cell types that were not represented in traditional scRNA-seq dataset [15]. Furthermore, the over-representation of stress response genes observed in scRNA-seq was not detected, while single-cell and single-nucleus platforms had equivalent gene detection sensitivity [13, 14, 16].

Here we report a robust method for preparing high-quality nuclei from various solid tissues from the woody plant genus *Populus* and demonstrate its use in single-nuclei transcriptome analysis. We show that this procedure results in intact mRNA isolation and limited leakage, with a broad representation of individual cell-types. To verify the performance of this method, we demonstrate this approach in tissues with contrasting properties, including *Populus* shoot apices from tissue culture grown plants containing mostly primary cell wall, and in lignified,

whole stems of *Populus* grown under greenhouse conditions. We further evaluated the repeatability of the single-nuclei transcriptome profiles derived from biological replicates.

## Methods

### Materials

- Forceps

- Razor blade

- Glass plate

- Aluminum heating/cooling block

- Eppendorf 5810R Refrigerated Centrifuge

- Rocking Platform Shaker

- Miracloth (Calbiochem, catalog number: 475855)

- Easystrainer 40 μM, for 50 ml tubes (Greiner Bio-One, catalog number: 542040)

- 2 ml Pasteur pipette

- Conical tube 50 ml (Fisherbrand, catalog number: 05-5390-6)

- 5 ml Test tube with caps (Falcon, catalog number: 352054)

- 1.5 ml RNase free non-stick Eppendorf low binding tubes (Thermo Fisher Scientific, catalog number: AM12450)

- RNaseZap wipes (Thermo Fisher Scientific, catalog number: AM9786)

- Rainin 1000 μl wide-bore sterile tips

- Rainin 200 μl wide-bore sterile tips

- Rainin 1000 μl pipet

- Rainin 200 μl pipet

- BD FACSAria IIU/III upgraded cell sorter

- 10% BSA Stock Solution (MACS BSA Stock Solution, Miltenyi Biotec, catalog number: 130-091-376)

- Protector RNase Inhibitor (Roche, catalog number: 3335402001)

- Spermine tetrahydrochloride (Sigma-Aldrich, catalog number: S1141-1G)

- Spermidine (Sigma-Aldrich, catalog number: S0266-1G)

- DTT (Thermo Fisher Scientific, catalog number: BP172-5)

- Nuclease-free water (Thermo Fisher Scientific, catalog number: AM9937)

- Nuclei Isolation Buffer (NIB) 4X (40 mM MES-KOH pH 5.7, 40 mM NaCl, 40 mM KCl, 12 mM $MgCl_2$, 10 mM EDTA, 1M sucrose). Prepare fresh NIB 1X with nuclease-free water and add 0.5 mM Spermidine, 0.1 mM Spermine, 1 mM DTT and 0.5 or 0.2 U/μl Protector RNase Inhibitor

- NIB WASH 4X (40 mM MES-KOH pH 5.7, 40 mM NaCl, 40 mM KCl, 12 mM MgCl$_2$, 1M sucrose). Prepare NIB WASH buffer 1X with nuclease-free water and add 0.1% BSA and 0.2 U/µl Protector RNase Inhibitor

## Plant material

We selected two plant material sources to develop and evaluate our nuclei isolation method, the *Populus* shoot apex and the whole stem. Both materials have different levels of cellular complexity and cell wall structure.

**Populus shoot apex.** Shoot tips and stem cuttings from *in vitro* grown hybrid *Populus* (*Populus tremula × alba* INRA clone 717 1B4) were used as explants for shoot multiplication. The explants were cut into small pieces (about 10–15 mm long) and placed aseptically on Murashige and Skoog media with vitamins (PhytoTech Labs, catalog number: M519) supplemented with 0.1 g/L myo-inositol (PhytoTech Labs, catalog number: I703), 0.25 g/L MES (PhytoTech Labs, catalog number: M825), 2% (w/v) sucrose (PhytoTech Labs, catalog number: S391) and solidified with 0.8% agar (PhytoTech Labs, catalog number: A296). The pH of the medium was adjusted to 5.8 with KOH before autoclaving. Explants were grown in a growth chamber under long day conditions (16 hours light/8 hours dark), 22˚C, 65% relative humidity, and 100 µmol m$^{-2}$ s$^{-1}$ photosynthetic photon flux. After three weeks in culture, a 5 mm-long portion of the shoot apex from 20 plants was excised, and leaves were removed, leaving only leaf primordia.

**Populus whole stem.** Rooted cuttings (clonal replicates) of the *Populus trichocarpa* reference genotype Nisqually-1 [17] were grown in a greenhouse with Peters Professional 20-20-20 Fertilizer (The Scotts Company, Marysvillle, OH) adjusted to 5mM with respect to nitrogen, in a flood irrigation system of ebb-and-flow benches. The ebb-and-flow benches were flooded once daily for approximately 30 minutes. Temperatures in the fan and pad cooled greenhouse ranged between 22˚C and 24˚C, and interior photosynthetically active radiation ranged up to 1200 µmol s$^{-1}$ m$^{-2}$ photosynthetic photon flux (over the waveband 400–700 nm) during the daily 14 hours of natural irradiance. After six weeks of growth, the whole stem (xylem, phloem and bark) internode interval between leaf plastochron index 4 and 5 was collected for nuclei isolation.

**Tissue preparation and nuclei isolation.** The following procedure describes the nuclei isolation from *Populus* shoot apices and stems, suitable for posterior processing using the 10× Genomics Chromium technology and single nuclei RNA sequencing (Fig 1). Before starting the nuclei isolation, all solutions, glassware, blades, 50 ml Conical Centrifuge Tubes, Eppendorf tubes, 2 ml Pasteur pipette, wide-bore tips, were precooled to 0–4˚C and kept on ice for the duration of the procedure. The glass plate was placed on an aluminum cooling block that was kept on ice. The entire procedure was executed in a cold room.

## Procedure

Twenty dissected shoot apices, or 1 single stem internode, were placed on a glass plate with 200 µl (shoot apices) or 400 µl (stem) of the Nuclei Isolation Buffer (NIB, 10 mM MES-KOH pH 5.7, 10 mM NaCl, 10 mM KCl, 3 mM MgCl$_2$, 2.5 mM EDTA, 250 mM sucrose, 0.1 mM spermine, 0.5 mM spermidine, 1 mM DTT, 0.5 U/µl Protector RNase Inhibitor). We modified the original buffer composition [18] to decrease nuclear membrane permeability and minimize RNA leakage (see *Nuclei Isolation Buffer (NIB)* in the Results section). Next, samples were chopped with a sterile razor blade for 2 minutes. This step was repeated twice (for the shoot apices) and three times (for the stem) with a 30 sec interval in between. The homogenate

**1.** Plant material: twenty dissected shoot apices, or 1 single stem internode.

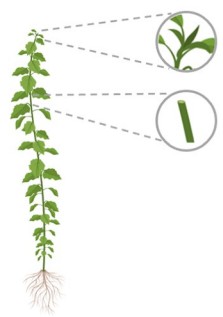

**2.** Chop on a glass plate with 200 μL (shoot apices) or 400 μL (stem) of NIB for 2 min. This step was repeated twice (shoot apices) and three times (stem) with a 30 sec interval in between.

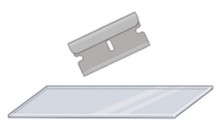

**3.** The homogenate was washed with 5 mL of NIB from the glass plate into a 50 mL conical tube, using a 2 mL Pasteur pipette, and incubated on a rocking shaker for 5 min.

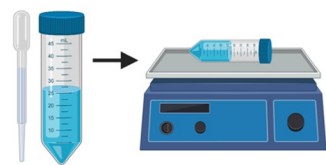

**4.** Filter homogenate through miracloth first, and then through 40 μm cell strainer, into a 50 mL conical tube. Centrifuge at 600 g for 5 min at 4°C.

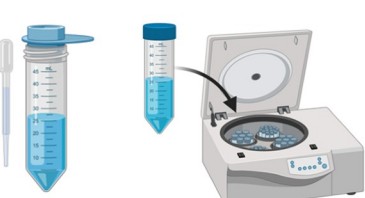

**5.** After centrifugation, the supernatant is carefully removed, and the pellet is resuspended in 4 mL of NIB WASH. Centrifuge at 600 g for 5 min at 4°C.

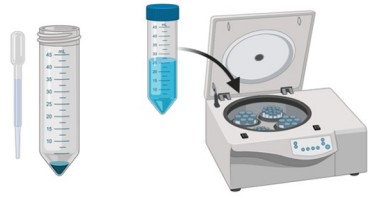

**6.** Step 5 is repeated for a total of two wash steps. After the second wash, the pellet is resuspended in 750 μL (shoots) to 1000 μL (stem) of NIB WASH.

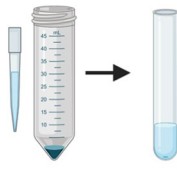

**7.** Nuclei are stained with 5 μg/mL DAPI, for 5 min at room temperature. Sort 40k nuclei using A BD FACSAria™ IIU/III upgraded cell sorter. *First quality checkpoint: overall nuclei integrity under confocal microscope (Fig. S1A)*

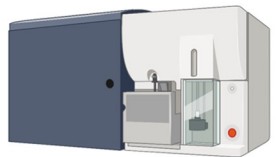

**8.** Load 20k nuclei into the 10 Genomics Chromium controller. Perform the cDNA synthesis following the 10× Genomics Chromium Single Cell v3.1 Dual Index Gene Expression protocol. *Second quality checkpoint: cDNA profile (Fig. S1B).*

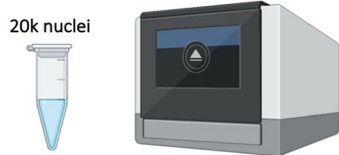

**9.** Single-nuclei RNA-seq library preparation. *Third quality checkpoint: library profile (Fig. S1C).*

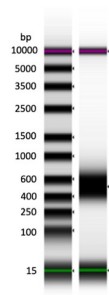

**Fig 1. Nuclei isolation workflow.** Workflow diagram showing the nuclei isolation and snRNA-seq library construction procedure from *Populus* shoots and stem.

was washed with 5 ml of NIB with lower Protector RNase Inhibitor concentration (NIB, 10 mM MES-KOH pH 5.7, 10 mM NaCl, 10 mM KCl, 3 mM MgCl$_2$, 2.5 mM EDTA, 250 mM sucrose, 0.1 mM spermine, 0.5 mM spermidine, 1 mM DTT, 0.2 U/μl Protector RNase Inhibitor) from the glass plate into a 50 ml conical tube, using a 2 ml Pasteur pipette, and incubated on a rocking shaker for 5 minutes of gentle horizontal shaking. Samples were then filtered through one layer of pre-wetted (using NIB) miracloth (Calbiochem) laid on top of a 50 ml conical tube placed on ice and tilted on its side. The filter was washed with 1–2 ml of NIB, to increase the nuclei recovered from the miracloth filter membrane. Next, samples were filtered through a pre-wetted (using NIB) 40 μm strainer (Greiner Bio-One) laid on top of a 50 ml conical tube placed on ice, tilted on its side and washed with an additional 1–2 ml of NIB. Samples were then centrifuged at 600 g for 5 minutes at 4˚C. After centrifugation, the supernatant was

carefully removed, and the pellet resuspended in 4 ml of NIB WASH (10 mM MES-KOH pH 5.7, 10 mM NaCl, 10 mM KCl, 3 mM MgCl$_2$, 250 mM sucrose, 0.1% BSA and 0.2 U/μl Protector RNase Inhibitor) pipetting very gently with a 2 ml Pasteur pipette. This step was repeated twice, except that after the last centrifugation the pellet was resuspended in 750 μl (shoots) to 1000 μl (stem) of NIB WASH and transferred to 5 ml test tubes used by the nuclei sorter (see *Fluorescence Activated Nuclei Sorting*). The sample's final nuclei concentration plays a significant role in the sample's behavior during the Fluorescence Activated Nuclei Sorting (FANS) step and it depends on the starting plant material amount and its nature. We recommend a final resuspension volume that allows sorting 40,000 nuclei in a relatively short period (10 to 20 minutes), enabling a final concentration of about 500 nuclei/μl after sorting.

The entire procedure, from chopping to loading 10× Genomics microfluidic chip with sorted nuclei, should be completed in less than 90 minutes to minimize RNA leakage and degradation which are the main issues compromising the success of snRNA-seq experiment.

## Assessment of nuclei integrity and concentration

Nuclei were stained with 5 μg/ml DAPI, for 5 minutes at room temperature. DAPI and nuclei were mixed by pipetting gently with the 1000 μl wide-bore sterile tips. To verify that the nuclei isolation procedure worked adequately, the overall nuclei integrity was analyzed using the laser excitation line of 405 and 488 in a Leica TCS SP5 confocal microscope, using 10 μl of the sample. The nuclei concentration was also estimated using a cell counting chamber with 10 μl of the sample (INCYTO C-Chip, catalog number: DHC-B02-5). Simultaneously, the remaining volume of the sample was used in the FANS step.

## Fluorescence Activated Nuclei Sorting (FANS)

To minimize the contamination of organelles such as chloroplast and remove the debris that could potentially clog the 10× Genomics microfluidic chips, we sorted the nuclei using FANS as described below.

A BD FACSAria™ IIU/III upgraded cell sorter with the following laser configuration (50 mw 488 blue; 100mw 350 UV; 50 mw 635 red; 100mw 405 violet and 100mw 531 yellow/green with 15 detectors and FSC/SSC) was used to sort 40,000 nuclei at a final concentration of approximately 500 nuclei/μl. The sheath tank was filled with autoclaved BD FACS Flow Sheath Buffer (BD Biosciences, catalog number: 342003). RNase inhibitor RibolockTM (Thermo Fisher Scientific, catalog number: EO0382) was added to each liter of sheath buffer to total of 10 L of sterile sheath padded to the ARIA per run. The Aria was then subjected to an aseptic sort set up within the Diva software and then manually cleaned with 10% bleach for 2 hours prior to samples being sorted. Next, the ARIA was cleaned with RNA/DNA non-DEPC treated nuclease free water (Thermo Fisher Scientific, catalog number: AM9932) for 30 minutes to rinse the system, followed by 30 minutes of sterile sheath buffer with 40U/μl of Ribolock per 12 ml sheath to wash away any potential RNAses from the internal components of the sorter. The sorter and all the surfaces of the computer and mouse were wiped down with RNaseZap wipes (Thermo Fisher Scientific, catalog number: AM9786) 30 minutes prior to sample acquisition. A70 integrated nozzle was used for the sorting of the nuclei. Upon completion of instrument preparation, CST and Accudrop calibrations, unstained nuclei were used for proper voltration of the DAPI signal, and to set the gating boundaries for the positive nuclei. Nuclei were sorted at an average rate of 226–350 ev/s with an abort rate of less than 5% and an efficiency on purity of over 97% recovery. Eight sorting tests were performed to optimize running conditions and dilution factors in order to get high recovery of intact nuclei with minimal membrane blebbing or leakage, on both post sort analysis and confocal microscopy

identification post sorting. This was done to maximize the total number of nuclei collected in under 80 μl of total sample volume. Collection was performed into 1.5 ml RNase free non-stick Eppendorf low binding tubes (Thermo Fisher Scientific, catalog number: AM12450) containing 10μl of NIB WASH. A total of 40,000 DAPI+ nuclei were sorted with a total recovery volume of 67–70 μl, at a speed of 1.0 flow rate. The speed at which samples were run directly affected the recovery quality and the nuclei leakage, thus a slower speed was optimized for better recovery of intact nuclei.

After the sorting, the nuclei integrity was examined under the scope as described above. After FANS, we also evaluate the absence of debris and organelles to determine the FANS' effectiveness.

## Generation of GEMs, cDNA synthesis and library preparation

A total of 20,000 nuclei were loaded into the 10× Genomics microfluidic chip, following the Single Cell v3.1 Dual Index Gene Expression protocol. NIB WASH was added to the 40 μl (500 nuclei/ μl) of nuclei suspension instead of water to reach the final volume required (43.2 μl). Then, the 31.8 μl of the cDNA master mix was slowly added to the nuclei suspension to minimize the osmotic shock. We added two additional PCR cycles during the cDNA amplification, for a total of 15 cycles. Finally, the snRNA-seq Dual Index library was constructed following the Single Cell v3.1 Dual Index Gene Expression protocol. We performed 12 cycles when using poplar apexes and 11 cycles for both biological replicates of poplar stem.

## Sequencing

Prior to full-scale sample sequencing, a preliminary run was performed in an Illumina iSeq (~ 4M reads) to allow the evaluation of parameters that are indicative of the quality of the single-cell RNA-seq library, including the estimated number of cells captured and the fraction of reads in cells (i.e., percentage of ambient or "leaked" RNA). The sequencing was performed using a standard program as follows: 28 bp (Cell barcode and UMI) for read 1, 90 bp (cDNA) for read 2, 10 bp for I7 Index, and 10 bp for I5 Index. The assessment of RNA leakage was derived from the profile of the relationship between UMI counts and cell barcodes detected. If the library was considered suitable for full-scale sequencing (reads in cells > 50% and at least 3,000 cells captured at low sequencing depth) we proceeded to sequencing in an Illumina NovaSeq6000. For full-scale sequencing we targeted 150-200M reads mapped to the genome per sample, which in our experience resulted in a sequencing saturation >70% and an average of 43,000 reads per nucleus.

## Data processing

Raw sequencing data were preprocessed using CellRanger (v 5.0). In brief, preliminary sequencing results (BCL files) were converted to FASTQ files using Cellranger mkfastq module by demultiplexing Chromium-prepared sequencing samples based on their barcodes. Then, the FASTQ files were aligned to v4.0 of the *Populus trichocarpa* reference genome (whole stem samples) or to v1.1 of *Populus tremula × alba* 717-1B4 reference genome (shoot apex samples), and UMI count data at a single-nucleus resolution was generated using Cellranger count module. To capture all the information in the pre-mRNA, we aligned the reads to a custom "pre-mRNA" reference that includes the introns information. The custom "pre-mRNA" references were built using the Cellranger mkref pipeline.

## Results

In the Methods section, we outlined a protocol of isolation of individual nuclei. This protocol was based in principle on a method described previously [18]. Here we describe the

optimizations made to this protocol for posterior sequencing of RNA extracted from nuclei. These optimizations had three primary goals: (1) minimize nuclei membrane damage to reduce mRNA leakage; (2) avoid mRNA degradation that results in a low-quality cDNA; (3) limit organelles' contamination, such as chloroplast, and remove cellular debris that interferes with microfluidic devices used in posterior sample preparation.

## Overall experimental conditions

All experimental procedures were carried out at 4˚C, in a walk-in cold chamber, to minimize mRNA degradation. Similarly, reagents, supplies, and instruments used in the nuclei isolation were cooled at least 30 min before use.

## Nuclei Isolation Buffer (NIB)

The composition of the NIB reported previously [18] was modified. Triton X-100, a non-ionic detergent commonly used to permeabilize cell membranes during nuclei isolation, was removed because it may contribute to RNA leakage [19, 20]. $MgCl_2$ (3mM) was included in all buffers to protect the nuclear membrane [21]. Finally, 0.2 U/μl of Protector RNase Inhibitor was added to reduce the activity of ribonucleases.

## Plant sample processing

Processing began with the chopping of the sample material in NIB for 2 min, with a 30 second rest interval between repetitions, until a homogenate solution was obtained. This step may have to be repeated multiple times depending on the plant material's hardiness. In the example provided here, this step was repeated twice for shoot apices, and three times for the stem samples. During this processing step, all the plant material must be in contact with the buffer containing RNA degradation inhibitors. However, excessive buffer needs to be avoided, as it reduces the efficiency of the chopping, affecting the consistency of the nuclei yield. We used 200 μl for shoot apices and 400 μl for the stem sample. Because RNA degradation can occur during this stage of sample preparation, a standard RNA extraction and quality evaluation (e.g., Agilent Bioanalyzer) can be performed post hoc to verify its integrity. If excessive degradation is observed, additional Protector RNase Inhibitor and DTT can be added to the NIB.

## Nuclei washes

After samples were filtered through miracloth and a 40 μm strainer, they were washed to remove ambient RNA from cellular cytoplasm or that may have leaked from nuclei during the isolation process. Standard protocols for plant nuclei isolation include centrifugation steps that range between 500–1500 g, for 5–10 min [18, 20, 22, 23]. In the nuclei washes we centrifuged samples at 600 g for 5 min, to minimize damage to nuclei while pelleting them at the bottom of the tubes. These parameters were optimal for nuclei isolated from *Populus* shoot apex and stem, but they may be adjusted for other plant species and tissues.

## Nuclei resuspension

The volume of NIB WASH buffer used to resuspend nuclei determines their concentration prior to FANS. Because nuclei yield is dependent on the species and tissue sampled, among other variables, the resuspension volume may need to be optimized according to the experimental conditions. Furthermore, the tissue source will determine the abundance of organelles and cell debris that will affect the performance of the cell sorter during the FANS. We adjusted the NIB WASH resuspension volume to allow sorting of approximately 40,000 nuclei (while

strictly adhering to the FANS parameters described in the Methods section), targeting a final concentration of approximately 500 nuclei/μl after sorting. This corresponded to a final resuspension volume of 750 μl for nuclei from shoot apices, and 1000 μl from stem.

In our procedure, the nuclei pellet was resuspended by gently pipetting NIB WASH buffer for 10 times, with a Rainin 1000 μl wide-bore sterile tip. In the event that nuclei and/or debris aggregates are still visible, we recommend additional pipetting to improve resuspension. However, pipetting should be minimized to limit nuclei damage and RNA leakage.

Following the nuclei resuspension, it is critical to evaluate the nuclei integrity under the confocal microscope. Well-preserved nuclei present well-defined borders, with no visible protrusion of nuclear content (S1A Fig).

## Fluorescence Activated Nuclei Sorting (FANS)

The FANS was incorporated to this protocol to remove cell debris and organelle contamination that may interfere with the microfluidic device used for processing individual nuclei (10× Genomics Controller). The FANS parameters described in the Methods section have been optimized to remove non-nuclear cellular debris and organelles while maintaining nuclei integrity (Fig 2). At an optimal final nuclei resuspension volume, FANS of approximately 40,000 nuclei (500 nuclei/μl) was completed in less than 10 min. If a longer sorting time is required to obtain this number of nuclei, it may indicate excessive aggregates and the need for additional pipetting during nuclei resuspension. Immediately after sorting, nuclei integrity and the presence of debris and organelles was examined under the confocal microscope, as described above, to determine the FANS' effectiveness (Fig 2 and S1A Fig). This quality control step must be performed just after FANS, taking only the amount required for visualizing the sample under the scope (5 to 10 μl). In parallel, the remaining sample can be prepared for the cDNA synthesis. Sample quality, including RNA leakage, can't be quantified at or beyond this point, and is only determined after the snRNA-seq library sequencing. In order to minimize mRNA leakage from nuclei and degradation, the procedure between the plant material collection and loading in the 10× Genomics microfluidic chip was completed in less than 90 min.

## cDNA and snRNA-seq library preparation

The procedures of cDNA synthesis and snRNA-seq library preparation followed the manufacturer's recommendations, with modifications to maximize cDNA synthesis and limit its degradation. 10× Genomics technology allows a range of nuclei to be loaded in the microfluidic chip. This will determine the final number of single-cell transcriptomes obtained. In our experiments we loaded 20,000 nuclei, targeting a final number of 10,000. NIB WASH was added to the 40 μl (500 nuclei/ μl) of nuclei suspension instead of water, up to the final volume required (43.2 μl). The NIB WASH composition was modified from NIB to make it compatible with the 10× cDNA synthesis reaction (see recipe in the Methods section). More specifically, EDTA, spermine and spermidine were removed from the original NIB WASH to reduce cDNA synthesis inhibition.

The cDNA master mix was slowly added to the nuclei suspension to minimize the rapid osmotic change. As nuclei contain less mRNA than whole-cells, two additional PCR cycles were added to the cDNA amplification step, for a total of 15 cycles. Suitable cDNA quality was confirmed by observing its profile in the Agilent D5000 High Sensitivity Tape Station (S1B Fig). Finally, the snRNA-seq Dual Index library was constructed following the Single Cell v3.1 Dual Index Gene Expression protocol. The number of library amplification cycles is dependent on the cDNA yield obtained after cDNA amplification. We performed 12 cycles when processing the *Populus* shoot apices and 11 cycles for both biological replicates of *Populus* stem

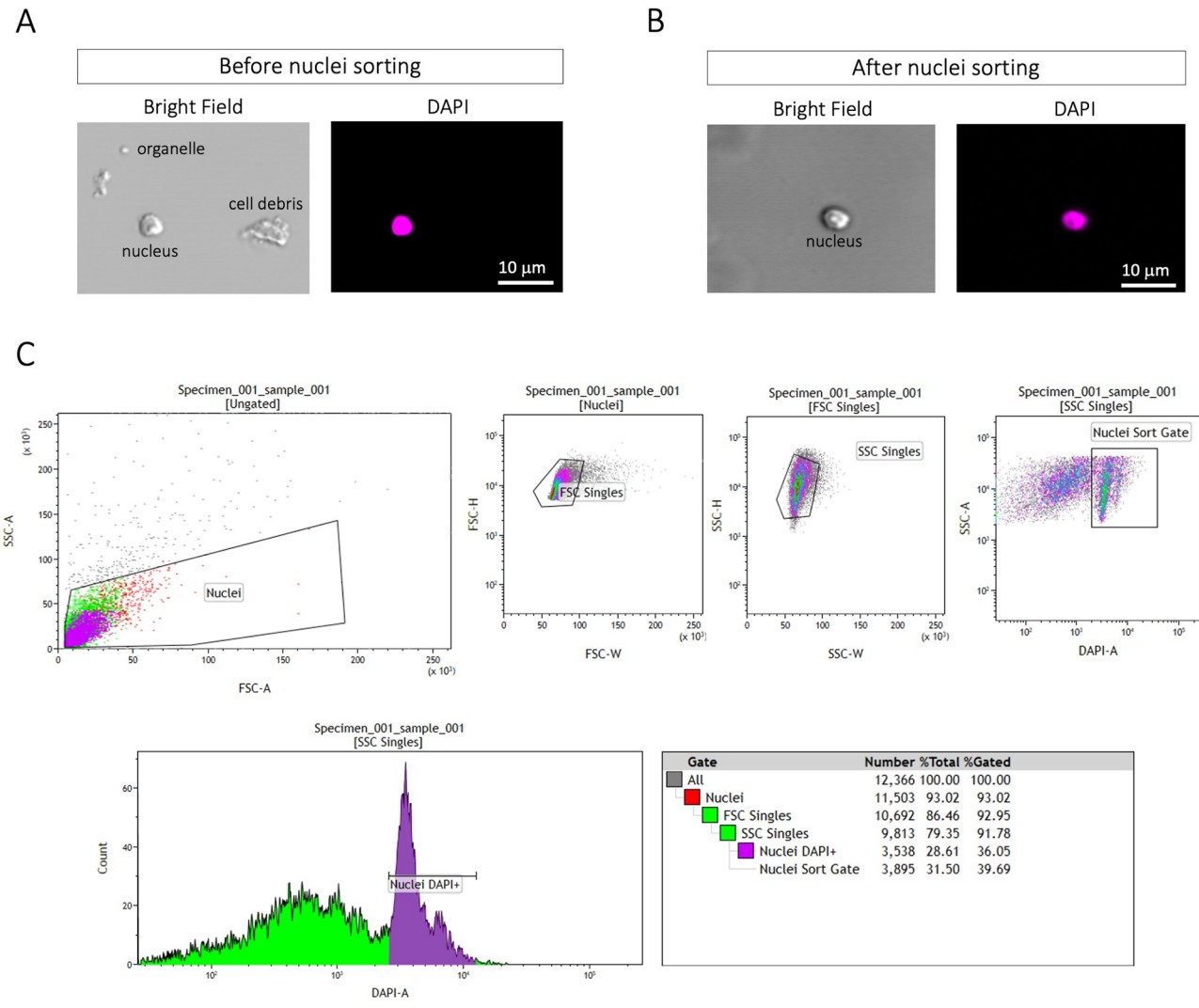

**Fig 2. Fluorescence Activated Nuclei Sorting. (A, B)** To minimize the contamination of organelles such as chloroplast and remove the debris that could clog the 10× Genomics microfluidic chips, we sort the nuclei using FANS technology using a BD FACSAria™ IIU/III upgraded cell sorter. **(C)** An initial gate was drawn on a FSC and SSC plot to eliminate the majority of non-nuclear debris and organelles. The final histogram was used to isolate the desired nuclei. 40,000 DAPI+ nuclei were sorted with a total recovery volume of 67–70 µl, into a 1.5 ml RNase free non-stick Eppendorf low binding tube containing 10 µl of NIB WASH.

samples. The snRNA-seq library profile evaluation using the Agilent Tape Station confirmed the high quality of the libraries used in the present work (S1C Fig).

## Single-cell profiling metrics

Approximately 43,000 reads were obtained per nucleus, of which over 73% mapped confidentially to the genome of reference (Table 1). We used the raw feature-barcode matrix to monitor data processing through all quality control steps, rather than the filtered feature-barcode matrix generated by Cell Ranger.

In the 10× Genomics microfluidic chip, reactions containing uniquely barcoded beads (referred hereafter as GEMs, for gel bead-in emulsions) and nuclei are partitioned into nanoliter droplets. However, most GEMs (90–99%) are typically not exposed and combined to a

**Table 1. Summary of snRNA sequencing and quality control overview for each sample.**

| Sample | SAM | Stem—Rep1 | Stem—Rep2 |
|---|---|---|---|
| Number of Nuclei | 9,430 | 7,383 | 8,245 |
| Mean Reads per Nucleus | 59,475 | 42,299 | 28,565 |
| Mean UMIs per Nucleus | 3,355 | 4,425 | 4,172 |
| Median UMIs per Nucleus | 2,975 | 3,722 | 3,328 |
| Mean Genes per Nucleus | 2,296 | 2,708 | 2,569 |
| Median Genes per Nucleus | 2,180 | 2,564 | 2,324 |
| Sequencing Saturation (%) | 75 | 74 | 79 |
| Reads Confidentially Mapped to the Genome (%) | 73 | 77 | 83 |
| Pearson's correlation between rep1 and rep2 | - | | 0.97 |

cell's/nucleus' RNA, but only ambient RNA. Therefore, we removed potentially empty GEMs (containing only ambient RNA) by requiring that a minimum of 1,000 unique molecular identifiers (UMI)-tagged transcripts be detected (Fig 3A and 3B). As shown in Fig 3, the inflection point (or 'knee') in the UMI cumulative plot is similar to the number of nuclei having at least 1,000 UMI-tagged transcripts (vertical dotted line). Using this quality control threshold, we determined the number of nuclei and the number of UMI-tagged transcripts per nucleus (Table 1). These UMI-tagged transcripts corresponded to the expression of an average of 2,296 to 2,708 genes per nucleus, for a total of 33,786 genes, 30,202 genes, and 30,408 genes detected for the shoot apices and stem (biological replicates 1 and 2) samples, respectively. Pearson's correlation between the normalized expression data of replicate 1 and replicate 2 is 0.97 (Table 1 and S2A Fig). Moreover, both replicates showed a high commonality between the expressed genes (S2B Fig).

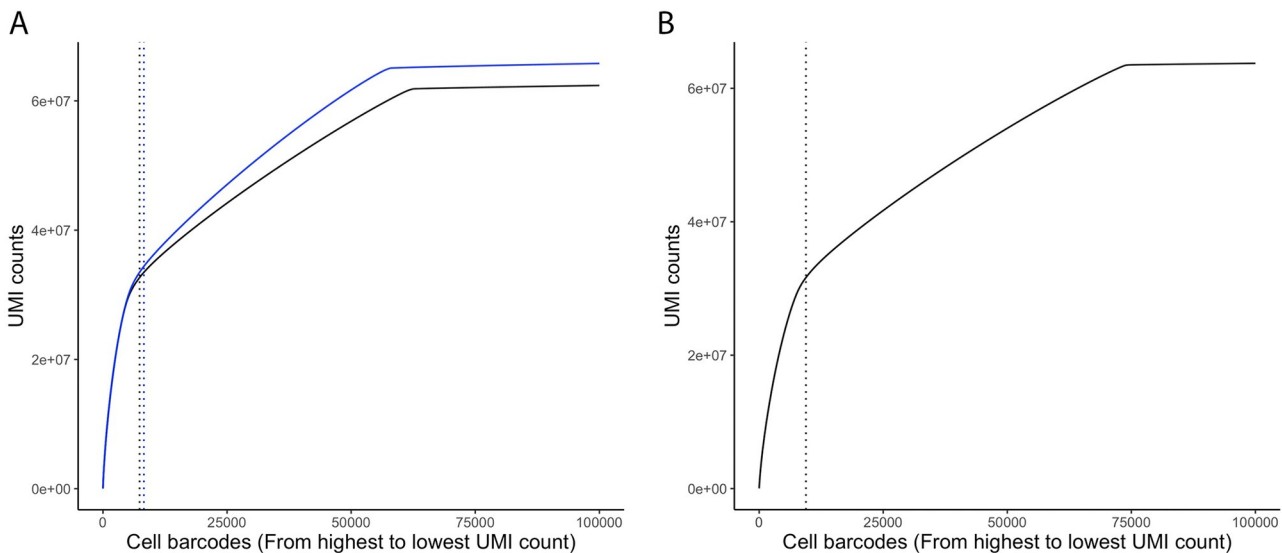

**Fig 3. Identification of true nucleus-associated barcodes.** Cumulative UMI counts plot—nucleus-associated barcodes, arranged in decreasing order (from highest to lowest UMI counts) versus the cumulative UMI counts. The vertical dotted lines indicate the 1,000 UMI-tagged transcripts cutoff. Nucleus barcodes with at least 1,000 UMIs were considered true nucleus. **(A)** Whole stem from biological replicate 1 and 2 (black and blue lines, respectively) of *Populus trichocarpa*. **(B)** Shoot apical meristem of *Populus tremula* × *alba* 717-1B4.

## Discussion

In the present work, we aimed to develop a nuclei isolation protocol for RNA sequencing, that overcomes the difficulties of generating protoplasts from solid plant tissues. Plant tissues' secondary cell wall hinders enzymatic digestion required for individual protoplast isolation, resulting in an unequal representation of cell types in a protoplast population. This limitation is especially critical for cell types located in the inner layers of a tissue or the inner tissues of an organ. While longer digestions may result in better dissociation of plant cells, the extended exposure to cell wall degrading enzymes used for protoplast isolation may cause stress responses that distort a cells' transcriptome profile. Furthermore, the process can damage the cellular membrane, leaking the mRNA cytoplasmic content. Thus, alternative approaches for transcriptome characterization of individual cells that bypass the need for enzymatic digestion and protoplast isolation are necessary. Initially, we tested three different nuclei isolation methods [18, 22, 23] for the plant material described in this work, which resulted in suboptimal nuclei integrity preservation [23], mRNA leaking [18], or high buffer stickiness during the FANS step [22]. Consequently, we developed the protocol described in this article.

To verify our nuclei isolation protocol's suitability for its application in single-cell transcriptome analysis, we prepared three snRNA-seq libraries using the 10× Genomics Chromium technology. One library was prepared using nuclei isolated from shoot apices from *Populus* tissue culture grown plants. Two libraries were prepared from a more lignified plant material, *Populus* stems from plants growing in the greenhouse. We targeted a final output of 10,000 single-cell transcriptomes following the 10× Genomics recommendations in all the cases. After sequencing, we obtained the transcriptomes of 9,430 nuclei for shoots and 7,383 and 8,245 for stems, which pointed to high predictability in the number of transcriptomes obtained when combining our protocol with the 10× Genomics Chromium system.

The fraction of the reads confidentially mapped to the genome was 73%, 77%, 83% for the Populus SAM sample, and for the stem sample replicates 1 and 2, respectively. The fraction of mapped reads given by Cell Ranger excludes those that map to two or more genome positions. A very recent whole-genome duplication (WGD) occurred in the genus *Populus*, resulting in nearly 8,000 pairs of paralogous genes of similar age (Tuskan et al. 2006). Due to the WGD, a certain percentage of reads mapped to regions of paralogous genes that may not have sufficiently diverged since the duplication event are likely to have been discarded. As expected, the percentage of the mapped reads is marginally lower in the *Populus tremula × alba* SAM sample, compared to *Populus trichocarpa* stem samples. For each gene of the 717-1B4 hybrid, the alleles of both *P. alba* and *P. tremula* are present. Still, the v1.1 of *P. tremula × alba* 717-1B4 reference genome contains a consensus sequence that does not distinguishes between both alleles. Thus, reads that originate from the *P. alba* or *P. tremula* genome segments could be discarded when mapped against the genome reference sequence, if they present a significant number of mismatched relative to the other species genome. Finally, the small differences in mapped reads from the two stem replicates could be explained by library preparation and sequencing bias.

Sequencing revealed an average of 2,296 expressed genes per nucleus for shoot apices and 2,708 (replicate 1) and 2,569 (replicate 2) for stems. Using similar microfluidics approaches, a significant variation in the number of expressed genes per nucleus was found in animal and human cells. For example, 553 genes per nucleus were found in pluripotent stem cells [24], 1,044 in retinal cells [25], 1,312 from human brain cells and 2,374 when studying isolated brain cells from mouse [26]. In plants, research that use isolated nuclei for microfluidics snRNA-seq methods remain very limited. Using the 10× Genomics Chromium system, an average of 1,126 expressed genes per nucleus were obtained using *Arabidopsis* roots [20, 22].

So far, most scRNA-seq analyses performed in plants used protoplasts from *Arabidopsis* plant tissues [9–12]. These papers also reported a significant variation in the number of genes per cell detected, ranging from 1,068 to 5,000. The sensitivity of detecting the single nucleus gene expression can be measured by the ratio between the number of UMI-tagged mRNA molecules captured in each nucleus and the number of detected genes per nucleus. This ratio represents the average number of mRNA molecules per gene captured per nucleus during the cDNA synthesis. The ratio was 1.3 for pluripotent stem cells [24], 1.49 for human brain nuclei, and 1.66 for mouse brain nuclei [26]. Very recently, a ratio of 1.40 was reported for a nuclei isolation protocol using Arabidopsis roots [27]. We found a ratio of 1.46, 1.63, and 1.62 for SAM, and stem replicates 1 and 2, respectively. Overall, our protocol arises as an alternative approach to generating a broad representation of individual cell transcriptomes for solid plant tissues when the generation of a sufficient and cell-type homogeneous protoplast population is challenging by enzymatic digestion.

## Supporting information

**S1 File.**
(DOCX)

**S1 Fig. Quality control checkpoints during nuclei isolation and library preparation. (A)** nuclei integrity is examined under the scope before and after nuclei sorting by FANS. After FANS, we also evaluate the absence of debris and organelles to determine the FANS' effectiveness. **(B)** cDNA yield and quality can only be determined after the cDNA amplification step. The good quality of the cDNA from *Populus* shoots and stem was confirmed by observing their profile in the Agilent Tape Station. **(C)** The final snRNA-seq libraries are also checked in the Agilent Tape Station to evaluate their quality.
(TIF)

**S2 Fig. Correlation between stem replicates. (A)** Correlation of the normalized gene expression between the two biological replicates performed using *Populus* stem samples. The high correlation between both biological replicates (R = 0.97) suggests a good reproducibility of the snRNA-seq when following the protocol developed in the present research work. **(B)** Venn diagram showing the commonality in the expressed genes between *Populus* stem snRNA-seq replicate 1 and replicate 2 data.
(TIF)

## Acknowledgments

We thank Egon Ranghini and Fred Souret, from 10× Genomics, for their technical support and advice in performing the methodology described in the present work.

## Author Contributions

**Conceptualization:** Daniel Conde, Paolo M. Triozzi, Henry W. Schmidt, Matias Kirst.

**Data curation:** Kelly M. Balmant.

**Methodology:** Daniel Conde, Paolo M. Triozzi, Andria L. Doty, Mariza Miranda, Anthony Boullosa.

**Resources:** Christopher Dervinis, Matias Kirst.

**Supervision:** Wendell J. Pereira, Christopher Dervinis, Matias Kirst.

**Writing – original draft:** Daniel Conde, Paolo M. Triozzi, Kelly M. Balmant, Andria L. Doty, Matias Kirst.

**Writing – review & editing:** Daniel Conde, Paolo M. Triozzi, Kelly M. Balmant, Henry W. Schmidt, Wendell J. Pereira, Christopher Dervinis, Matias Kirst.

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
