## [Decision Letter · Decision Letter 0]

9 Apr 2021

PONE-D-21-04833

A robust method of nuclei isolation for single-cell RNA sequencing of solid tissues from the plant genus Populus

PLOS ONE

Dear Dr. Kirst,

Thank you for submitting your manuscript to PLOS ONE. After careful consideration, we feel that it has merit but does not fully meet PLOS ONE’s publication criteria as it currently stands. Therefore, we invite you to submit a revised version of the manuscript that addresses the points raised during the review process.

We look forward to receiving your revised manuscript.

Kind regards,

Ram Kumar Sharma, Ph.D

Academic Editor

PLOS ONE

Additional Editor Comments:

There are minor revision to be responded.

Journal Requirements:

Reviewers' comments:

Reviewer's Responses to Questions

**Comments to the Author**

1. Is the manuscript technically sound, and do the data support the conclusions?

Reviewer #1: Yes

Reviewer #2: Yes

2. Has the statistical analysis been performed appropriately and rigorously? 

Reviewer #1: Yes

Reviewer #2: Yes

3. Have the authors made all data underlying the findings in their manuscript fully available?

Reviewer #1: Yes

Reviewer #2: No

4. Is the manuscript presented in an intelligible fashion and written in standard English?

Reviewer #1: Yes

Reviewer #2: Yes

5. Review Comments to the Author

Reviewer #1: General comments-

Conde et al. has proposed an extended method for nuclei preparation from complex plant samples for single cell sequencing analysis which is key challenge for plant researchers. The introduction is very concise and well composed and I must appreciate it. The optimized protocol is very nicely descried in method sections that supports its possible application for single cell RNA-Seq analysis in other complex plants. Results and discussion are well written in context to key precautions and challenges associated with nuclei isolation and downstream single cell sequencing. The extended methodology proposed in this research paper will facilitate further developments of scRNA-Seq analysis in many other important plant systems. Overall, the current manuscript satisfies all the necessary points to be published in Plos ONE, hence I recommend it for publication in this journal.

Some minor corrections/ comments

Comment 1: Correct “can be only be” to “can only be” in S1 file line 76.

Reviewer #2: The manuscript by Conde et al, entitled “A robust method of nuclei isolation for single-cell RNA sequencing of solid tissues from the plant genus Populus”, seems to be interesting, wherein authors have developed an alternating approach for generating high-quality protoplasts from plant tissue by characterizing the mRNA extracted from individual nuclei instead of whole cells. They have tested the protocols in two different plant materials with varying cellular complexity levels and cell-wall structure, Populus shoot apices, and more lignified stems. The manuscript is well written with clear objective highlighted in the study, and alternative protocol developed by authors will be useful for performing single cell transcriptome analysis.

Line#375 – 376: Reason for 73% genome mapping rate should be discussed in the discussion section. There is a significant difference in the genome mapping rate of reads on stem-Rep1 and Stem-Rep2.

Line#392 – 395: “These UMI-tagged transcripts corresponded to the expression of an average of 2,296 to 2,708 genes per nucleus, for a total of 33,786 genes, 30,202 genes, and 30,408 genes detected for the shoot apices and stem (biological replicates 1 and 2) samples, respectively”. What is the gene specific correlation between bio rep 1 and 2 in this study?

Line#428 – 429: “2,708 (replicate 1) and 2,569 (replicate 2) for stems”, what is the commonality in number of genes between both the replicates. It would be better to represent this data in Venn diagram.

The authors should also provide and discuss the significance in the expression level of genes in comparison to the previously available protocols.

6. PLOS authors have the option to publish the peer review history of their article (what does this mean?). If published, this will include your full peer review and any attached files.

Reviewer #1: No

Reviewer #2: No

---

## [Author Response · Author response to Decision Letter 0]

15 Apr 2021

We are pleased to submit to PLOS ONE a revised version of our manuscript PONE-D-21-04833, entitled: “A robust method of nuclei isolation for single-cell RNA sequencing of solid tissues from the plant genus Populus”. We would like to thank the PLOS ONE Editorial Board and the two anonymous reviewers for their helpful comments and suggestions, which helped improve the manuscript.

In this revised version we incorporated the reviewers’ comments and suggestions. Below you will find point-by-point answers to the reviewers’ comments, which are indicated in italic and bold. We also included a version of the manuscript with track changes.

We would like to certify that this manuscript has not been published and is not under consideration for publication elsewhere. We do hope that the revised manuscript will satisfy reviewers’ expectations.

Sincerely,

Matias Kirst

 

Reviewer: 1

Comments to the Author 

Conde et al. has proposed an extended method for nuclei preparation from complex plant samples for single cell sequencing analysis which is key challenge for plant researchers. The introduction is very concise and well composed and I must appreciate it. The optimized protocol is very nicely descried in method sections that supports its possible application for single cell RNA-Seq analysis in other complex plants. Results and discussion are well written in context to key precautions and challenges associated with nuclei isolation and downstream single cell sequencing. The extended methodology proposed in this research paper will facilitate further developments of scRNA-Seq analysis in many other important plant systems. Overall, the current manuscript satisfies all the necessary points to be published in Plos ONE, hence I recommend it for publication in this journal.

Minor Concerns: 

• Comment 1: Correct “can be only be” to “can only be” in S1 file line 76.

This modification was made in the new version of the manuscript.

 

Reviewer: 2 

Comments to the Author 

The manuscript by Conde et al, entitled “A robust method of nuclei isolation for single-cell RNA sequencing of solid tissues from the plant genus Populus”, seems to be interesting, wherein authors have developed an alternating approach for generating high-quality protoplasts from plant tissue by characterizing the mRNA extracted from individual nuclei instead of whole cells. They have tested the protocols in two different plant materials with varying cellular complexity levels and cell-wall structure, Populus shoot apices, and more lignified stems. The manuscript is well written with clear objective highlighted in the study, and alternative protocol developed by authors will be useful for performing single cell transcriptome analysis.

Concerns: 

• Line #375 – 376: Reason for 73% genome mapping rate should be discussed in the discussion section. There is a significant difference in the genome mapping rate of reads on stem-Rep1 and Stem-Rep2.

We have incorporated comments about the mapping rate in the Discussion component of the manuscript, as described below: 

“The fraction of the reads confidentially mapped to the genome was 73 %, 77 %, 83 % for the Populus SAM sample, and from the stem sample replicates 1 and 2, respectively. The fraction of mapped reads given by Cell Ranger excludes those that map to two or more genome positions. A very recent whole-genome duplication (WGD) occurred in the genus Populus, resulting in nearly 8,000 pairs of paralogous genes of similar age (Tuskan et al. 2006). Due to the WGD, a certain percentage of reads mapped to regions of paralogous genes that may not have sufficiently diverged since the duplication event, and were discarded from further analysis. As expected, the percentage of the mapped reads is marginally lower in the Populus tremula × alba SAM sample, compared to Populus trichocarpa stem samples. For each gene of the 717-1B4 hybrid, the alleles of both P. alba and P. tremula are present. Still, the v1.1 of Populus tremula × alba 717-1B4 reference genome contains a consensus sequence that does not distinguishes between both alleles. Thus, reads that originate from the P. alba or P. tremula genome segments could be discarded when mapped against the genome reference sequence, if they present a significant number of mismatched relative to the other species genome. Finally, the small differences in mapped reads from the two stem replicates could be explained by library preparation and sequencing bias.” 

• Line #392 – 395: “These UMI-tagged transcripts corresponded to the expression of an average of 2,296 to 2,708 genes per nucleus, for a total of 33,786 genes, 30,202 genes, and 30,408 genes detected for the shoot apices and stem (biological replicates 1 and 2) samples, respectively”. What is the gene specific correlation between bio rep 1 and 2 in this study?

The Pearson’s correlation value between the normalized expression data of replicate 1 and replicate 2 was added to Table 1 of the manuscript. Moreover, a graph showing the correlation between normalized gene expression between replicates 1 and 2 was added to Figure S2A. Accordingly, the following sentence was added to the results section:

“Pearson’s correlation between the normalized expression data of replicate 1 and replicate 2 is 0.97 (Table 1 and Fig S2A).”

• Line #428 – 429: “2,708 (replicate 1) and 2,569 (replicate 2) for stems”, what is the commonality in number of genes between both the replicates. It would be better to represent this data in Venn diagram.

A Venn diagram showing the commonality in the expressed genes between Populus stem snRNA-seq replicate 1 and replicate 2 was added to Figure S2B. Accordingly, the following sentence was added to the results section: “Moreover, both replicates showed a high commonality between the expressed genes (Fig S2B).”

• The authors should also provide and discuss the significance in the expression level of genes in comparison to the previously available protocols.

Our understanding is that the reviewer is concerned about the sensitivity of detection of gene expression in single nuclei when using this approach with the 10× Genomics Chromium technology. This sensitivity can be measured by the ratio between the number of UMI-tagged mRNA molecules captured in each nucleus and the number of genes detected per nucleus. This ratio represents the average number of mRNA molecules per gene captured per nucleus during the cDNA synthesis. Accordingly, we added to the discussion section the following paragraph where we compare the ratio obtained using our protocol, with previously published protocols published for animal and plant nuclei, when this data is available:

“The sensitivity of detecting the single nucleus gene expression can be measured by the ratio between the number of UMI-tagged mRNA molecules captured in each nucleus and the number of detected genes per nucleus. This ratio represents the average number of mRNA molecules per gene captured per nucleus during the cDNA synthesis. The ratio was 1.3 for pluripotent stem cells [24], 1.49 for human brain nuclei, and 1.66 for mouse brain nuclei [26]. Very recently, a ratio of 1.40 was reported for a nuclei isolation protocol using Arabidopsis roots [27]. We observed a ratio of 1.46, 1.63, and 1.62 for SAM, and stem replicates 1 and 2, respectively.”

---

## [Editor Report · Decision Letter 1]

21 Apr 2021

A robust method of nuclei isolation for single-cell RNA sequencing of solid tissues from the plant genus Populus

PONE-D-21-04833R1

Dear Dr. Kirst,

We’re pleased to inform you that your manuscript has been judged scientifically suitable for publication and will be formally accepted for publication once it meets all outstanding technical requirements.

Kind regards,

Ram Kumar Sharma, Ph.D

Academic Editor

PLOS ONE

Additional Editor Comments (optional):

NA
---

## [Editor Report · Acceptance letter]

30 Apr 2021

PONE-D-21-04833R1 

A robust method of nuclei isolation for single-cell RNA sequencing of solid tissues from the plant genus *Populus*

Dear Dr. Kirst:

I'm pleased to inform you that your manuscript has been deemed suitable for publication in PLOS ONE. Congratulations! Your manuscript is now with our production department. 

Kind regards, 

on behalf of

Dr. Ram Kumar Sharma 

Academic Editor

PLOS ONE